# Phosphate Enrichment of Niobium-Based Catalytic Surfaces in Relation to Reactions of Carbohydrate Biomass Conversion: The Case Studies of Inulin Hydrolysis and Fructose Dehydration

**Mariana N. Catrinck** [1,2,†], **Sebastiano Campisi** [1,†], **Paolo Carniti** [1], **Reinaldo F. Teófilo** [2], **Filippo Bossola** [3] **and Antonella Gervasini** [1,*]

1 Dipartimento di Chimica, Università degli Studi di Milano, Via Camillo Golgi 19, 20133 Milano, Italy; mncatrinck@gmail.com (M.N.C.); sebastiano.campisi@unimi.it (S.C.); paolo.carniti@unimi.it (P.C.)
2 Departamento de Química, Universidade Federal de Viçosa, Viçosa 36570-900, Brazil; rteofilo@ufv.br
3 Consiglio Nazionale delle Ricerche—Istituto di Scienze e Tecnologie Chimiche "Giulio Natta"—(CNRSCITEC), Via Camillo Golgi 19, 20133 Milano, Italy; filippo.bossola@scitec.cnr.it
\* Correspondence: antonella.gervasini@unimi.it; Tel.: +39-0250314254
† These authors equally contributed to the paper.

**Abstract:** In this work, some physical mixtures of $Nb_2O_5 \cdot nH_2O$ and $NbOPO_4$ were prepared to study the role of phosphate groups in the total acidity of samples and in two reactions involving carbohydrate biomass: hydrolysis of polyfructane and dehydration of fructose/glucose to 5-hydroxymethylfurfural (HMF). The acid and catalytic properties of the mixtures were dominated by the phosphate group enrichment. Lewis and Brønsted acid sites were detected by FT-IR experiments with pyridine adsorption/desorption under dry and wet conditions. Lewis acidity decreased with NbP in the composition, while total acidity of the samples, measured by titrations with phenylethylamine in cyclohexane (~3.5 µeq m$^{-2}$) and water (~2.7 µeq m$^{-2}$), maintained almost the same values. Inulin conversion took advantage of the presence of surfaces rich in Brønsted sites, and $NbOPO_4$ showed the best hydrolysis activity with glucose/fructose formation. The catalyst with a more phosphated surface showed less deactivation during the dehydration of fructose/glucose into HMF.

**Keywords:** niobium-based catalysts; surface acidity; inulin hydrolysis; fructose dehydration; 5-hydroxymethylfurfural; catalyst deactivation

## 1. Introduction

Niobium (Nb)-based materials have been widely investigated in the past decades as promising and versatile heterogeneous catalysts [1–4]. In the class of niobium compounds, niobic acid (i.e., partly hydrated niobium oxide, $Nb_2O_5 \cdot nH_2O$, hereafter labelled as NbO) [5,6] and niobium oxophosphate ($NbOPO_4 \cdot nH_2O$, hereafter labelled as NbP) [7,8] have aroused considerable interest as active phases, as supports, and also as promoters. In any case, these materials have consolidated their success mainly as solid acid catalysts [9,10] due to their acid strength (Hammett acidity function $-8.2 < H_o < -5.6$ for NbO [4] and $-8.2 < H_o < -3.0$ for NbP [11]), co-presence of both Lewis and Brønsted acid sites (LAS and BAS, respectively), and capability to preserve Lewis acidity even in the presence of water. Indeed, in both NbO and NbP, the acidity in water is maintained thanks to the formation of $NbO_4$-$H_2O$ adducts, which in part still exhibit Lewis acid character, as proved by Nakajima through Raman and infrared spectroscopy studies of CO adsorption [12]. Actually, taking advantage of this combination of BAS and water-tolerant LAS, NbO and NbP have found successful exploitation in polysaccharidic biomass processing, in particular for hydrolysis and dehydration reactions [13–24], albeit with some limitations in terms of resistance to deactivation. Indeed, in these reactions, NbP may easily undergo leaching of phosphate

groups, in particular in hot water [25], while NbO, although more stable in water than NbP, may be blocked by insoluble products (humins or coke) whose deposition at the catalytic surface seems to be promoted by medium-to-strong Lewis acid sites [22].

A fine control of the acidity [26] in terms of nature (LAS/BAS), strength, and surface density of sites is then required to improve the stability of these catalysts as well as to address the intrinsic complexity of reaction pathway when highly functionalized molecules are involved, as in the case of polysaccharide valorization processes [27–35].

Several strategies, such as surface treatment [36–38], metal doping [30,39–41], structural distortions [42,43], and mixed oxide formation [44–50] have been proposed over the years for controlling the acidity of Nb-based catalysts. It is regrettable that most of these approaches imply complicated synthetical routes or limited reproducibility. It is also known that the ratio of phosphorus to metal represents a key parameter for tuning the surface properties, and in particular surface acidity, of metal phosphate catalysts [51–54]. On the other hand, the synthesis of Nb-based catalysts with controlled phosphorus content is not an easy task. A more practical and cheaper alternative may consist in making balanced mixing of NbO and NbP to create physical mixtures with controlled P content to attain the desired goal in terms of LAS-to-BAS ratio, acid site number, and acid strength. Recently, a physical mixture obtained by a combination of equal weighed amounts of NbO and NbP was reported to exhibit interesting catalytic performances compared to the separated and pure NbO and NbP solids in the reaction of glucose dehydration to 5-hydroxymethylfurfural (HMF) reaction [55].

Starting from these encouraging results, in this, work NbO and NbP have been mechanically mixed in different proportions to obtain solids with increasing P content. Afterwards, physical mixtures have been characterized with particular focus on the acidic features: the total number and weak-to-strong acid site ratios were determined by solid–liquid phase titrations by phenylethylamine probe adsorption, while pyridine adsorption, monitored by infrared spectroscopy, allowed for discriminating and quantifying LAS and BAS.

The expected differences in the acidic features of the NbO:NbP mixtures in terms of distribution of the nature, number, and strength of acid sites might influence the catalytic performances. To prove this, two different reactions have been studied: inulin hydrolysis and fructose/glucose dehydration to HMF, both being reactions sensitive to the nature and strength of the acid sites on the surface of Nb-based materials.

## 2. Results

### 2.1. Mixture Preparation and Characterization

Niobic acid (NbO) and niobium oxophosphate (NbP) powder samples, supplied from CBMM, have the composition reported in detail in the Experimental section. The morphological properties of the samples have been already studied and reported in the literature [55]; both NbO and NbP samples were mesoporous solids with specific surface areas of 177 and 140 $m^2 \ g^{-1}$, respectively. Starting from these two components, three physical mixtures were prepared with NbO/NbP mass ratios of 1:3, 1:1, and 3:1, by accurately weighing precise aliquots of the individual materials and intimately mixing them through ball milling.

The intended compositions (expressed as wt%) of the physical mixtures are reported in Table 1, with the sample codes used hereafter. In addition, considering the aim of the work and the central role of phosphate groups in the following discussion, samples have been classified according to their normalized phosphorous content, calculated by normalizing the P-concentration in each sample to the one of NbP, considered as the reference. In doing so, NbO and NbP represent the lowest and uppermost extremes of the series, with normalized P-content values equal to zero and one, respectively; for example, the NbO: NbP mixture (3:1) contains a quarter of phosphorous of NbP (Table 1).

**Table 1.** Composition of niobium-based catalysts.

| Code | Nominal Composition (wt%) | | | | Normalized P Content |
|------|------|------|------|------|------|
| | **Nb** | **P** | **O** | **K** [a] | |
| NbO | 69.9 | - | 30.1 | - | 0 |
| NbO:NbP(3:1) | 66.0 | 2.1 | 31.3 | 0.5 | 0.25 |
| NbO:NbP(1:1) | 62.3 | 4.2 | 32.5 | 1.1 | 0.5 |
| NbO:NbP(1:3) | 58.6 | 6.2 | 33.6 | 1.6 | 0.75 |
| NbP | 55.0 | 8.2 | 34.7 | 2.1 | 1 |

[a] impurity deriving from $K_2O$ presence in NbP powder.

The compositional uniformity of powder mixtures was evaluated by EDX micro-analysis, by mapping out the areal distribution of elements and comparing it with the elemental composition measured at single points. The good agreement between punctual and areal composition values (Table S1) confirmed that all the prepared mixtures had uniform composition.

Semi-quantitative information from EDX analyses were also used to roughly estimate the P content of the samples. Plotting phosphorous concentration from EDX analysis as a function of the normalized P content, an almost linearly increasing trend is obtained, although, in all cases, P content from EDX estimations was higher than the corresponding nominal values (black empty symbols in Figure 1).

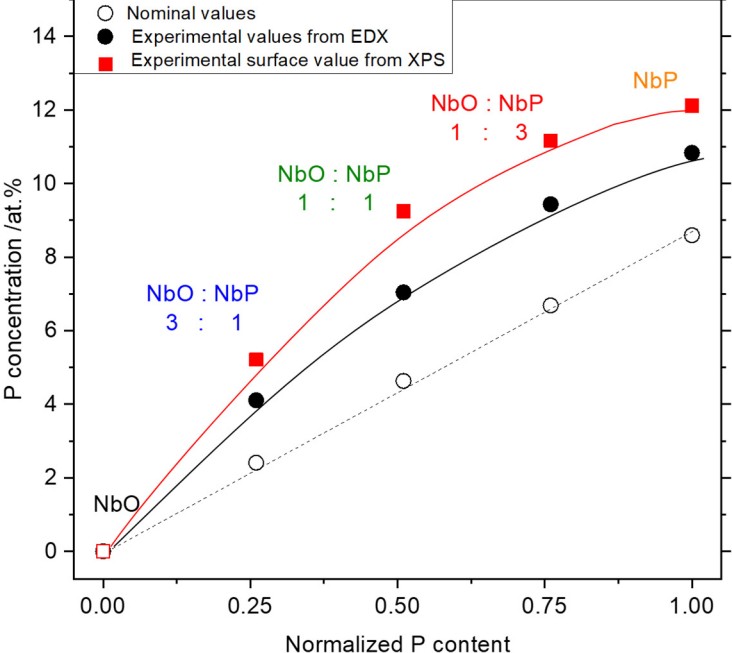

**Figure 1.** Phosphorous concentration of the NbO and NbP samples and relevant physical mixtures as a function of normalized P content (calculated assuming unit value, *p* = 1, for NbP). Nominal values refer to the expected composition based on the weighed amount of NbO and NbP in the NbO:NbP mixtures and sample compositions as indicated in Table 1.

The observed trend seems to suggest a phosphate surface enrichment, then, characterization by X-ray photoelectron spectroscopy (XPS) was performed to determine in a more appropriate way phosphorous surface concentration (Table S1). As expected, the experimental values of P-concentration obtained by XPS analyses indicated a higher surface concentration of phosphorous than that calculated from nominal compositions of samples (Table 1), which assume a homogeneous phosphorous presence from the surface to the bulk of the samples. This indicated that, for the NbP preparation, phosphating was likely carried out on finite niobium oxide. Additionally, the XPS values were higher than the EDX

ones since EDX technique is characterized by larger interaction volume and lower surface sensitivity than XPS. Thus, from the XPS analysis, a clear gradual surface enrichment of phosphorous could be argued within the sample series, passing from NbO to NbP.

The high-resolution spectra of the main elements (Nb, O, and P) for all the analyzed samples are shown in Figure 2, and the surface concentrations for the same elements are reported in Table S1. The Nb 3d photoelectron signal of the NbO:NbP(1:1) and NbO:NbP(3:1) mixtures appears as a spin–orbit doublet that can be resolved into the $3d_{5/2}$ and $3d_{3/2}$ components. The binding energy (BE) position of these two components (nearly 207.6 eV and 210.4 eV for $Nb3d_{5/2}$ and Nb $3d_{3/2}$, respectively) as well as the multiplet splitting values (ca. 2.7–2.8 eV) and the intensity ratio of the two peaks (about 2:3) are typical of $Nb^{5+}$ species in bulk $Nb_2O_5$.

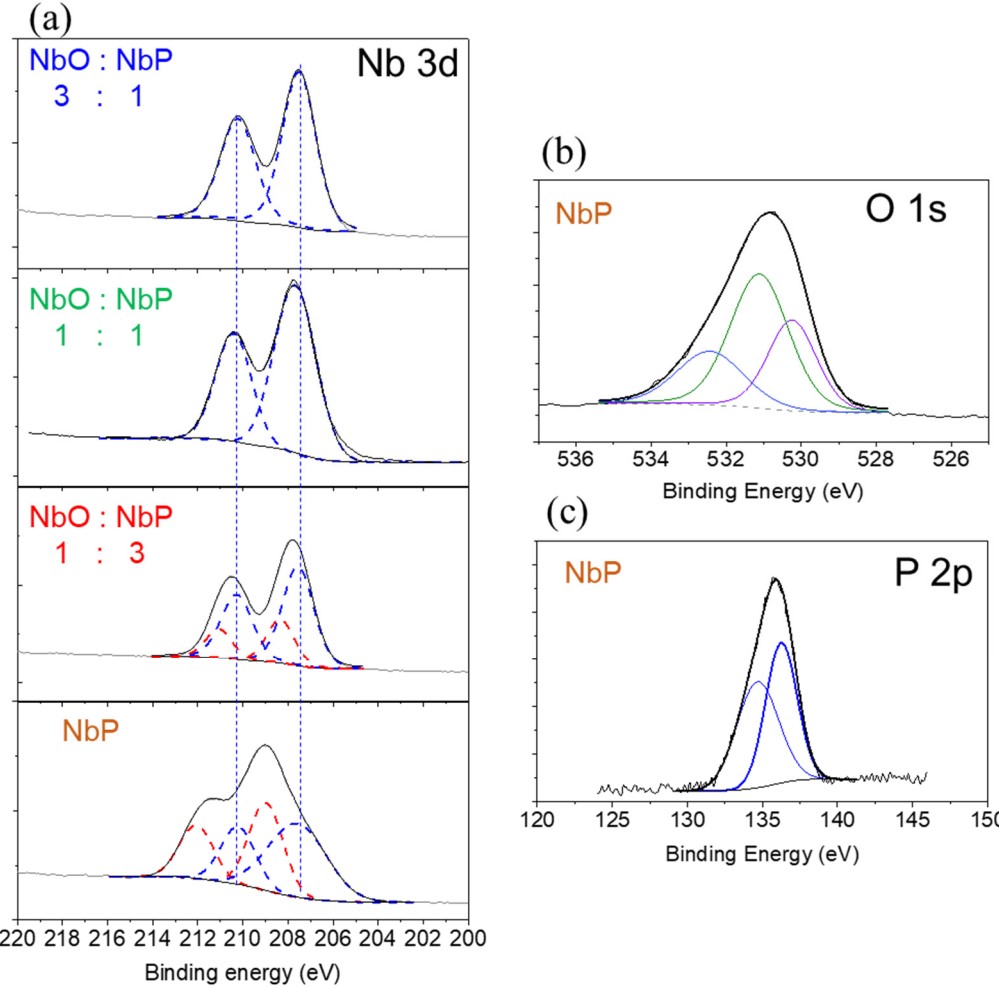

**Figure 2.** Surface composition analysis by XPS: high resolution spectra of: (**a**) Nb 3d region for NbP and mixtures, (**b**) O 1s region, and (**c**) P 2p region for NbP.

The Nb 3d spectra recorded on NbO:NbP(1:3) and NbP samples are broader than those recorded on NbO:NbP(1:1) and NbO:NbP(3:1) and can be decomposed into two different doublets. One doublet has components centred at the same BE as the already-observed ones (ca. 207.6 and 210.4 eV, dotted blue curves), and it can be then ascribed to $Nb^{5+}$ in $Nb_2O_5$, while the second one has sub-peaks with BE values ca. 0.8–1.0 eV higher than those observed for $Nb_2O_5$ (ca. 208.5 and 211.2 eV, dotted red curves). According to the literature, this contribution is associated with a highly ionic $Nb^{5+}$ species and it can be ascribed to $Nb^{5+}$ involved in P–O–Nb–O bonds [56,57]. In fact, this second contribution was predominant in the NbP sample, and it gradually lowered and disappeared by decreasing P content in the samples. Oxygen signal in the 1s region was fitted with three components,

located at 530.2, 531.1, and 532.4 eV associated with oxygen involved in the Nb–O and Nb–O–P bonds and in phosphate groups (O–P) [25]. Finally, the P 2p spectrum can be decomposed as a doublet having a spin-orbital split of ca. 0.9 eV and centred at 134.6 eV, characteristic of typical phosphate groups [25].

## 2.2. Characterization of Surface Acidity

### 2.2.1. Intrinsic and Effective Acidity by Liquid–Solid Phase Titrations

The acidity of the surfaces of NbO and NbP and their ability to maintain acidity in water, for the most part, are well-known properties starting from the pioneering works of Ziolek (among others, [2,4]) and Okuhara [58]. They opened the possibility of using some solids containing niobium as acid catalysts in heterogeneous reactions in water, thanks to the acidic water-tolerance properties possessed by these solids.

To prove both the intrinsic and effective acidities of the studied samples, namely NbO, NbP, and their mixtures, acid–base titrations by using phenyl ethylamine (PEA), as a basic probe, in cyclohexane and water:isopropanol solution (80:20 $v/v$, corresponding to the solvent used in the studied fructose/glucose dehydration reaction), respectively, were carried out. The results obtained are reported in Table S2 as total density of the acid sites with the percentage of strong acid sites. The results have been obtained, starting from the collected PEA adsorption isotherms measured at 30 °C that are shown in Figure 3. On each sample, two successive isotherms were collected, on the fresh sample (I° isotherm) and on the PEA-saturated sample (II° isotherm) after it was purged with pure solvent overnight to remove weakly adsorbed PEA. By this procedure, the amount of total, weak, and strong acid sites (corresponding to $PEA_{ads,max}$, obtained by the Langmuir model equation used to fit the isotherms) of each sample could be quantified. Then, considering the value of the surface area of each sample, the acid site density (equivalent of acid site per unit sample surface, µequiv m$^{-2}$, Table S2) could be computed.

Regarding the intrinsic acidity measured in cyclohexane (Figure 3, left side), all the samples adsorbed a high approximately similar amount of the PEA probe, except NbP, which adsorbed a little less PEA, probably due to the lower surface area value than NbO. The strength of the acid sites of the surfaces was quite strong; strong acid sites corresponded to ca. 66–73% of the titrated total sites without any clear trend with the sample composition. The acidity picture of the samples changed slightly when the titrations were carried out in polar-protic solvent; the relative adsorption isotherms of PEA in water:isopropanol solution (Figure 3, right side) gave rise to the effective acid site density of the samples that are shown in Table S2. Based on the tolerant properties of the two samples NbO and NbP [59], a maintenance of acidity was expected by all the samples, including the NbO: NbP mixtures. In water:isopropanol solution, the amount of PEA adsorbed by NbO decreased by ca. one third compared to the same measured in cyclohexane, while NbP adsorbed PEA without any decrease. In general, a decrease of the strong acid sites was observed for all the samples compared with the intrinsic strong acid sites. The ratios between the total effective and intrinsic acid sites (E.A/I.A.) were 0.67 for NbO and 1 for NbP and around 0.7 for the NbO:NbP mixtures. From all the obtained results, the good acidity of all the surfaces and their water-tolerant properties clearly emerges, with NbP appearing as the best acid surface in the polar–protic medium.

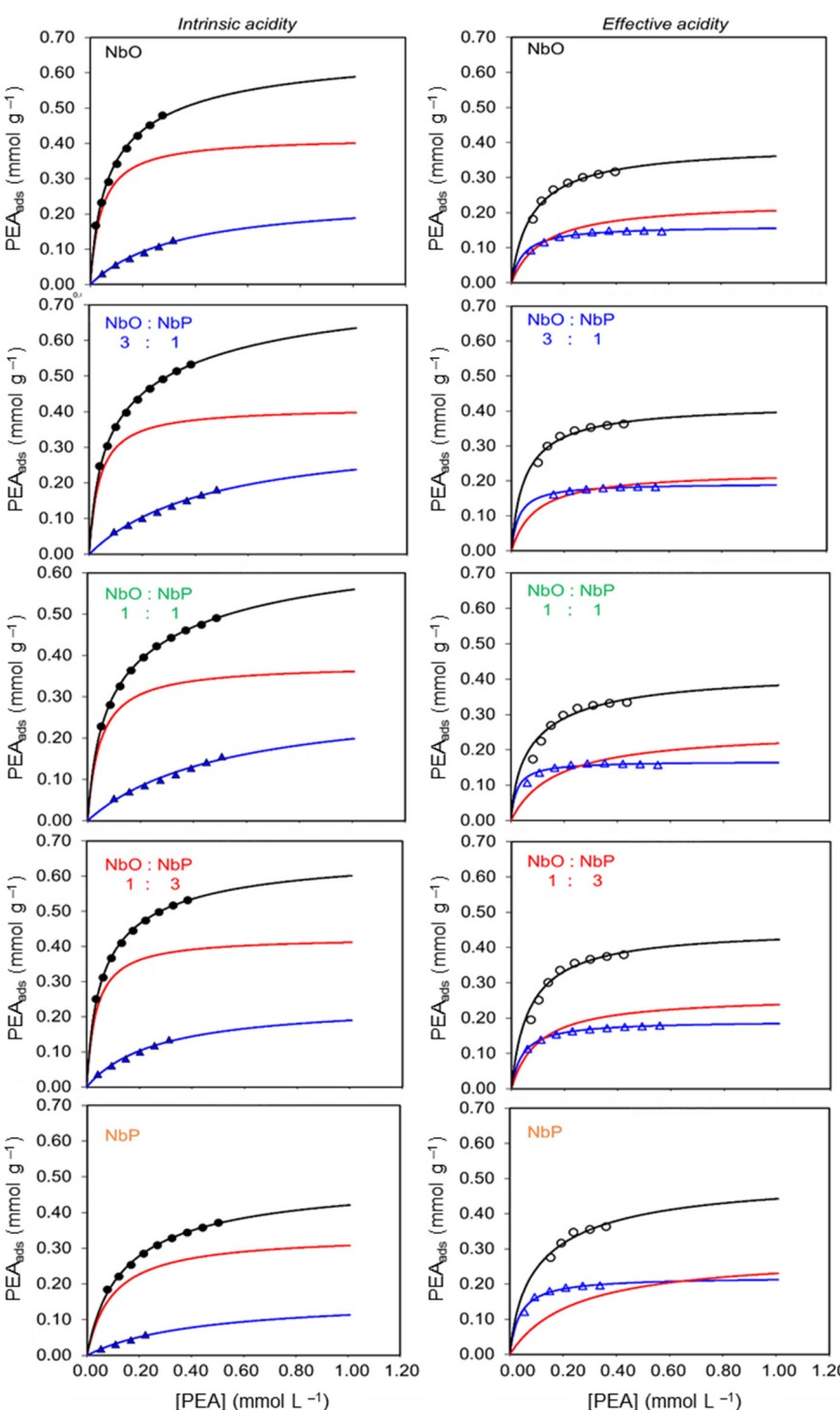

**Figure 3.** Adsorption isotherms at 30 °C of PEA probe on Nb-based samples evaluated in cyclohexane (left, full markers) for measuring the *intrinsic* acidity and in water-isopropanol solution (80:20 $v/v$), (right, empty markers) for the *effective* acidity. Calculated Langmuir curves of I° run PEA adsorption (total acid sites), black lines; calculated Langmuir curves of the II° run PEA adsorption (weak acid sites), blue lines; calculated Langmuir of the strong acid sites, obtained from difference from the I° and II° PEA adsorptions, red curves.

### 2.2.2. Acid Site Intrinsic and Effective Nature

Besides the amount of sites and acid strength, the nature of the acid sites, as well as their tolerance to water, is of great importance to a comprehensive knowledge of the acid

properties of any surface. Through FT-IR spectroscopic analyses of pyridine adsorption–desorption, LAS can be easily distinguished from BAS, studying the nature of the formed molecular complexes after adsorption/desorption of suitable base probes. Samples of NbO, NbP, and their mixtures were contacted with pyridine both in the vapor phase and aqueous solution and FT-IR desorption spectra at 150 °C in the range of 1560 to 1400 cm$^{-1}$ are shown in Figure S1.

All spectra exhibited the same bands pattern typical of formation of pyridine complexes with BAS and LAS types but with different intensities, indicating that the relative densities of the acid sites differed considerably among the samples. In particular, the sharp band at 1445 cm$^{-1}$, due to the interaction of pyridine with LAS ($\nu_{19b}$ mode), and the broad band at 1540 cm$^{-1}$, assigned to the pyridinium ion bounded to BAS ($\nu_{19b}$ mode), were observed on all the samples, while the band at 1488 cm$^{-1}$ ($\nu_{19a}$ mode) is normally related to the simultaneous interaction of pyridine coupled with BAS and LAS [31,60].

As expected, the FT-IR pyridine desorption spectra collected under dry conditions (Figure S1a) showed more intense bands than those recorded in the presence of water (Figure S1b). It suggested a decrease in both LAS and BAS acid sites on the surfaces when water was present, even if limited. Then, experiments carried out with pyridine under dry conditions allowed determining the *intrinsic* nature of the acid sites, which means the number LAS and BAS characteristic of the surfaces. On the other side, experiments performed using an aqueous solution of pyridine investigated on the *effective* nature of the acid sites, which means the amount of LAS and BAS in real working conditions (i.e., in aqueous solutions). Both NbO and NbP are known to possess water-tolerant Lewis acid sites on their surface that have been associated with NbO$_4$ species in tetrahedral coordination [43]. Brønsted acidity has been related to Nb-OH groups on NbO, while on NbP, both P-OH and Nb-OH groups concur to the acidity; the former is considered a slightly stronger Brønsted acid site than Nb-OH [13,43].

The FT-IR results collected with vapor pyridine and pyridine solution (Figure S1 and Table S2) indicated that NbO and NbP exposed a similar total amount (LAS *plus* BAS) of acid sites (0.132 meq g$^{-1}$ for NbO and 0.115 meq g$^{-1}$ for NbP). However, the distribution of LAS and BAS varied on the two samples and then on the related mixtures formed from the two acid solids. NbP possessed the higher amount of BAS, which decreased within the mixtures by increasing NbO amount, while the amount of LAS was higher for pure NbO and smaller for pure NbP.

The amount of LAS and BAS and the LAS/BAS ratio determined in dry and wet conditions have been correlated to the surface P-concentration of each sample (as determined by XPS, Table S1). The trends observed in the two conditions differ in some amounts. In general, a decrease in the amount of *intrinsic* LAS as a function of normalized surface P-concentration was observed, while a similar amount of *effective* LAS was observed on all the samples. *Intrinsic* and *effective* BAS increased with an exponential trend against normalized surface P-concentration. The *intrinsic* LAS to BAS ratio decreased almost linearly from pure NbO to pure NbP (Figure 4a), and the *effective* LAS to BAS ratio decreased in an exponential way when water was concerned (Figure 4b) since LAS suffer more from the presence of water than BAS. These observations suggest that the amounts of LAS and BAS can be tunable by varying the percentage of NbO and NbP in the sample mixture.

By comparing the acid site populations of the samples determined with and without water (Figure 4a vs. Figure 4b), it can be found that the amounts of LAS and BAS on NbO had reductions 57% and 45%, respectively, when water was concerned in comparison with the experiment performed with pyridine in the vapor phase. On the contrary, NbP showed a slighter decrease (only 15% of LAS and 19% of BAS), confirming the superior water-tolerant acid properties.

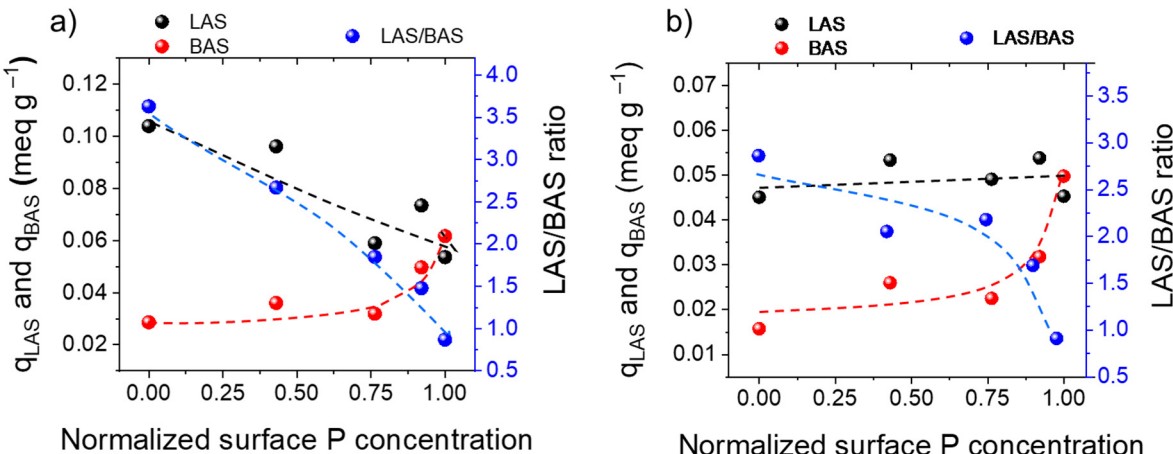

**Figure 4.** Trends of the amount of LAS and BAS ($q_{LAS}$ and $q_{BAS}$) and LAS/BAS ratio against normalized surface P-concentration (obtained from XPS) determined following pyridine desorption at 150 °C in experiments with pyridine vapor (*intrinsic* acidity) (**a**); and pyridine aqueous (*effective* acidity) (**b**).

The decrease, even if slight, in the number of both LAS and BAS in the presence of water can be explained by the competition between water molecules and pyridine probe for the adsorption on the acid sites. However, the different impact of water on the LAS and BAS population among the studied samples can be rationalized considering that the Brønsted acidity increases with the increment of P in the samples, and the P-OH groups lead to a slightly stronger Brønsted acidity than Nb-OH sites [2,13,43]. The environment surrounding NbO$_4$ tetrahedra can affect its water-tolerant activity [43]. The presence of the P-OH groups on the equatorial plane of the NbO$_6$ octahedron [7] might cause a sort of steric effect of protection of the LAS.

### 2.3. Catalytic Activity

2.3.1. Catalytic Inulin Hydrolysis

Inulin is a polysaccharide naturally occurring in plants from the Asteraceae family (such as chicory, Jerusalem artichoke, barley, garlic, onion, rye, wheat, and other roots and tubers), where it acts as a means of energy storage. From a chemical point of view, inulin is a polydisperse mixture of *β*-(2,1) fructans, and polymeric linear chains whose backbones are made up by fructose units (connected by *β*-(2,1)-d-fructosyl-fructose bonds) and terminated by a glucose unit (linked by an *α*-d-glucopyranosyl bond). Depending on the source, different distributions of degree of polymerisation (DP) can be observed, with values ranging from 2 to 60. Thus, inulin is usually labelled with the general formula GF$_n$, where G stands for glucose, F for fructose, and n for the average DP [60–62].

Inulin can undergo depolymerization in solution through enzymatic or acid hydrolysis of the β-(2,1) bonds, thus resulting in a fructose-rich syrup, which is a suitable feedstock to produce several interesting platform molecules. A schematic representation of the reaction of inulin hydrolysis and of production of reducing sugars is shown in Scheme 1.

Mineral acids and protonic solid acids with significant BAS population have been studied as active acid catalysts for inulin depolymerisation [17,48]. In particular, low LAS/BAS ratios and high effective acid strength in water are responsible for superior activity in breaking the β-(2,1) bonds of inulin chains.

The five studied samples with different P-concentrations were then tested as catalysts in the inulin hydrolysis reaction carried out in water.

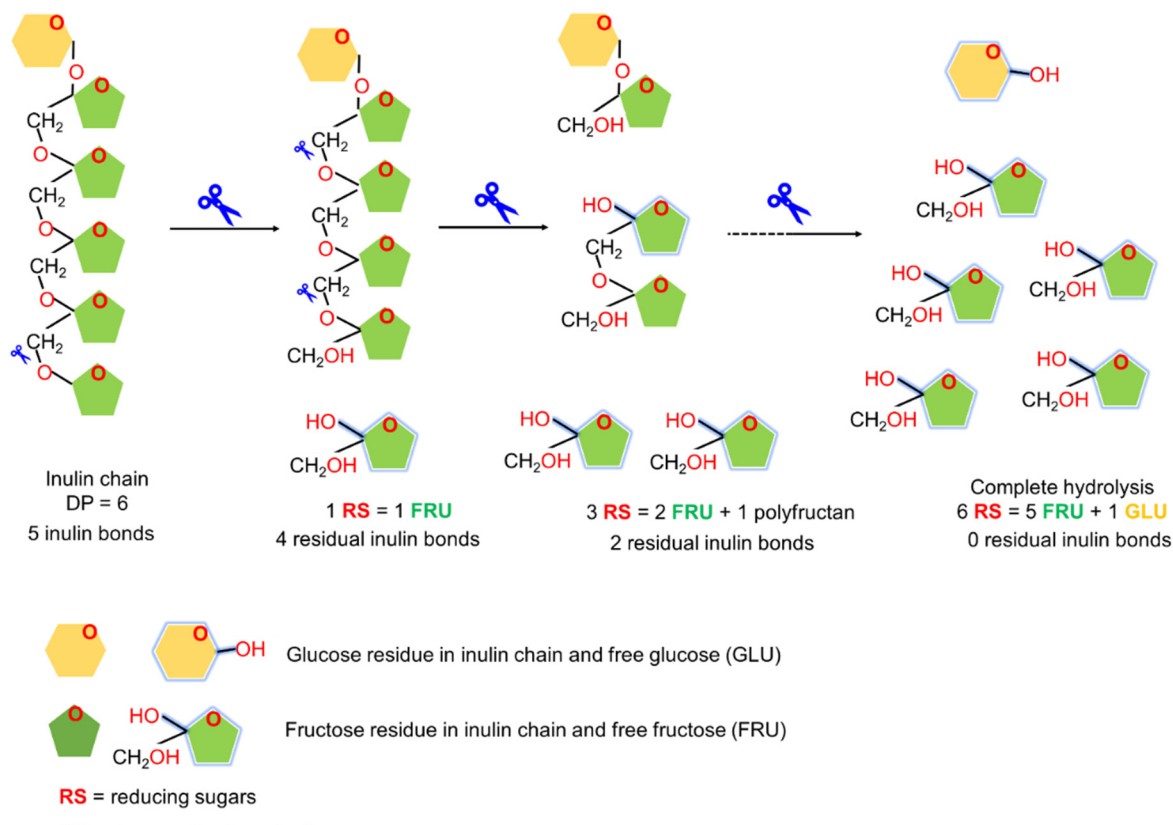

**Scheme 1.** Schematic representation of inulin hydrolysis reaction and production of reducing sugars (fructose, glucose, and polyfructans). Although terminal units are more accessible and more reactive, acid-catalysed hydrolysis of inulin chains can proceed with random scission of the interior or terminal bonds. In the former case, a monomer and a polyfructan chain reduced by one unit are formed, while in the latter case, two shortened polyfructan chains are obtained. The ultimate step of the hydrolysis process should consist of all the inulin being completely converted to fructose and glucose (the monomers). Whatever the location of the bond broken during the process, inulin hydrolysis is accompanied by a gradual increase in the amount of total reducing sugars (fructose, glucose, and polyfructans, all contain an anomeric carbon and thus can be classified as reducing sugars).

The progress of the inulin hydrolysis reaction over all the studied samples was followed as a function of time/temperature in the 50−90 °C interval operating in batch-working mode. According to the reaction pathway (depicted in Scheme 1), the catalytic activity was evaluated as the capability to hydrolyse inulin to total reducing sugars (RS, Figure S2) as well as to form monosaccharides (fructose and glucose, F + G).

Figure 5 shows the obtained results on NbO and NbP as well as on the NbO:NbP mixtures in terms of (i) the number of residual inulin bonds, which decreased with reaction temperature (Figure 5a), and (ii) the formation of total reducing sugars (Figure 5b), which increased with reaction temperature. The progress of the reaction is associated with the increase in the reaction temperature. Inulin hydrolysis started at ca. 70 °C in the presence of NbP and the NbO:NbP samples with normalized P content ≥ 0.5 (i.e., NbO:NbP(1:1), NbO:NbP(1:3), and NbP). For these three samples, experimental curves are almost superposed until 80 °C, while for higher temperature, the conversion curves tend to diverge along exponential trends. In particular, as expected, the higher the P content, the higher the inulin conversion (i.e., lower number of residual inulin bonds and higher concentration of formed reducing sugars were detected).

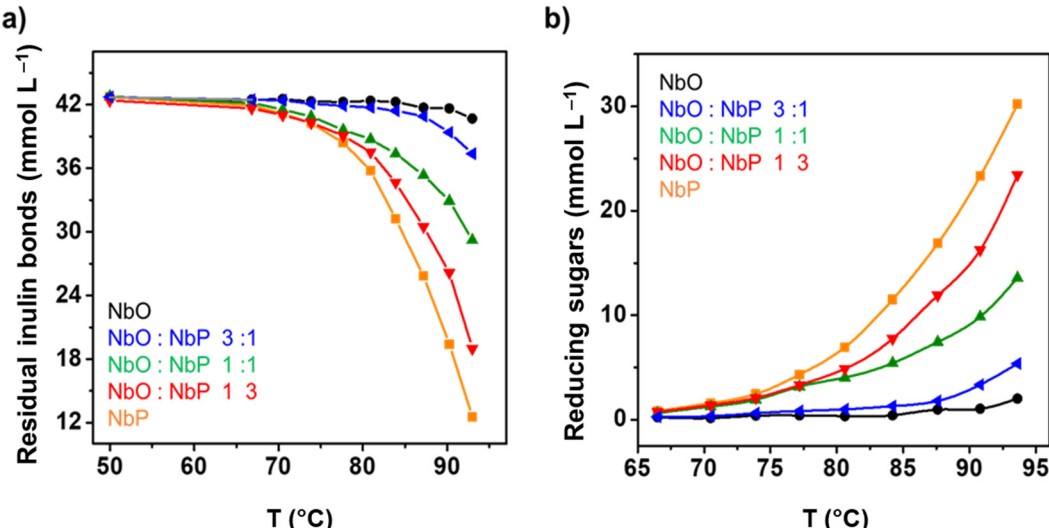

**Figure 5.** Results of the batch catalytic reaction of inulin hydrolysis in water on NbO, NbP, and relevant NbO:NbP mixtures as a function of reaction temperature (from 50 to 90 °C with linear increase of 0.12 °C min$^{-1}$), expressed as residual inulin bond concentration (**a**); and reducing sugar concentration (**b**).

Differently, pure NbO and the NbO:NbP samples containing the lowest normalized P content (P = 0.25; NbO:NbP(3:1)) became active only above 80 °C, and just a modest conversion was attained at 90 °C. These results confirm that the inulin depolymerization over Nb-based catalysts is determined by the Brønsted character of the surface, since the progressive phosphate enrichment on the surfaces of Nb-based catalysts leads to ever more active catalysts. This observation is further validated by the exponentially increasing trend, shown in Figure 6a, that reports inulin conversion at 90 °C, expressed as the ratio between the reducing sugars produced and the reducing sugars at complete inulin hydrolysis, as a function of the normalized surface P concentration, obtained from XPS analysis. Concerning product distribution at 90 °C (Figure 6b), the ratio between polyfructans and monosaccharides produced by the hydrolysis was much higher for the less active NbO and NbO:NbP(3:1) samples (which also possess low P content), consistently with the low activity of these catalysts.

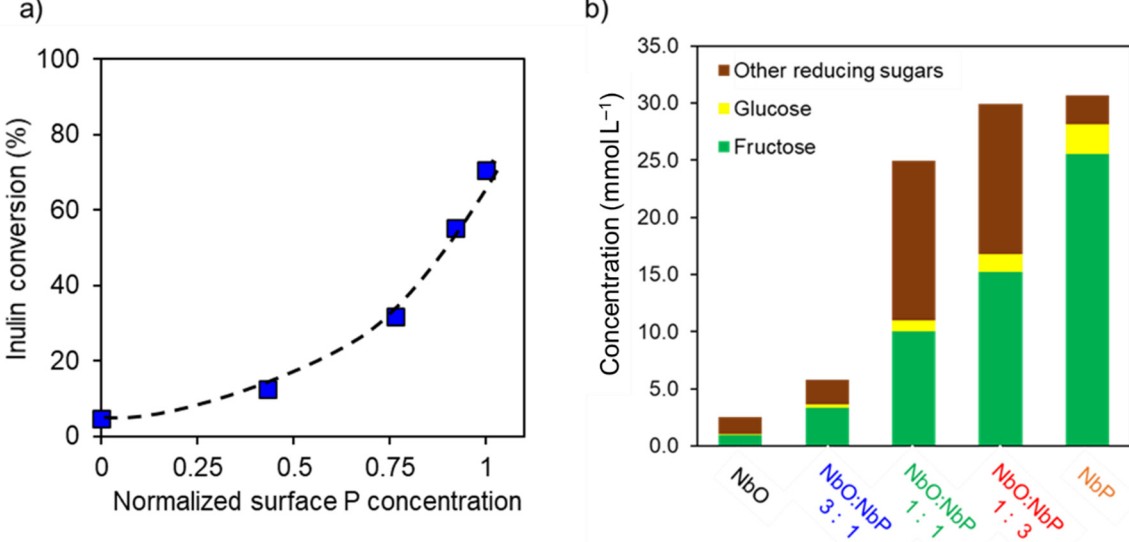

**Figure 6.** Catalytic performances of pure NbO and NbP and relevant mixtures in the reaction of hydrolysis of inulin evaluated at 90 °C; inulin conversion against normalized surface P concentration of samples (**a**); and distribution of the reducing sugar concentration (glucose, fructose, and polyfructans) for each sample (**b**).

The kinetic coefficients of the inulin hydrolysis were calculated at different average temperatures (average temperature $T_m$ values were calculated at time intervals of ca. 30 min corresponding to ca. $\Delta T = 5\,°C$, according to the temperature program) under the assumption of first order reaction; the results at three representative reaction temperatures (70, 80, and 90 °C) are listed in Table 2 together with the activation parameters ($E_a$ and lnA), determined from the Arrhenius plot. By comparing kinetic constant values, it appears clear that the activity significantly increases passing from NbO to samples with gradually increasing phosphorous content. The discrepancy between samples with low P content and P-rich samples is more pronounced at a high temperature, the kinetic constants values at 90 °C ($k_{90}$) of NbO and NbO:NbP (3:1) samples being almost two orders lower than the other catalysts. No significant differences emerged in the computed activation parameters for the five catalysts ($E_a$ around 100–140 kJ mol$^{-1}$ and ln A around 30–45). In any case, the obtained values for the individual components, NbO and NbP, were in good agreement with those previously reported in the literature [17,48].

**Table 2.** Kinetic and activation parameters for the reaction catalytic hydrolysis of inulin.

| Catalyst | Inulin Hydrolysis [a] | | | | |
|---|---|---|---|---|---|
| | $k_T$ [b] (h$^{-1}$) | | | $E_a$ (kJ mol$^{-1}$) | *ln*A (A *in* h$^{-1}$) |
| | 70 °C | 80 °C | 90 °C | | |
| NbO | 0.00304 | 0.00457 | 0.00387 | n.d [b] | n.d. [b] |
| NbO:NbP(3:1) | 0.00304 | 0.00771 | 0.0763 | 107.1 ± 20.1 | 32.6 ± 6.9 |
| NbO:NbP(1:1) | 0.00721 | 0.0625 | 0.142 | 99.8 ± 8.8 | 31.1 ± 3.0 |
| NbO:NbP(1:3) | 0.0312 | 0.0808 | 0.307 | 123.9 ± 6.5 | 40.1 ± 2.2 |
| NbP | 0.0373 | 0.141 | 0.571 | 136 ± 5.3 | 44.6 ± 1.8 |

[a] catalyst mass used, ca. 0.3 g; [b] too low conversion obtained.

## 2.3.2. Catalytic Dehydration of Fructose/Glucose

The dehydration of polysaccharidic biomass to 5-hydroxymethyl-2-furaldehyde (HMF) through heterogeneous acid catalysis is considered a key reaction for a sustainable production of platform chemicals. Indeed, HMF can undergo several transformations to produce added-value compounds, such as monomers, solvents, fuels, additives, and intermediates to fine chemicals [63].

When acid solids are used as catalysts in the fructose dehydration reaction, both BAS and LAS are active sites to form HMF. However, the control of surface acidity represents a crucial tool for maximizing the selectivity/yield to HMF, which can suffer from the interference by unwanted side reactions. A marked Lewis acidity seems to promote side-condensation and side-polymerization reactions, giving rise to the so-called *humins*, with parallel decreased selectivity to HMF and pronounced activity declining. Actually, being highly insoluble in water, *humins* tend to deposit onto the catalyst surface, causing deactivation. From this point of view, the choice of an optimal reaction solvent represents a key point for this reaction. The use of solvents, apolar liquids (such as dimethyl sulfoxide), or hydroalcoholic mixtures able to limit/prevent *humin* formation or to favour/enhance their solubilisation has been pursued by several researchers [64,65]. Besides the formation of *humins* under acid conditions, HMF rehydration to levulinic acid (LA) can also occur, contributing to a decrease in the selectivity to HMF. This consecutive reaction has been proven to be suppressed carrying out the reaction under flow conditions in a tubular reactor [66].

Among the different types of acid solid catalysts used for the fructose dehydration, NbO and NbP have both received great attention in the literature due to their ability to effectively convert fructose to HMF [9,13,22]. In this study, NbO and NbP and two selected physical mixtures (NbO:NbP 3:1 and NbO:NbP 1:1) were tested in a fructose dehydration reaction to evaluate the effect of phosphate concentration on the activity, selectivity, and catalytic stability. To minimize the extent of side reactions, catalytic tests were carried

out under flow conditions and by using a water–isopropanol (80:20 *v/v*) mixture as the reaction solvent, with the final intent to limit the *humin* deposition on the catalyst surface and to enhance the catalyst stability. A fructose and glucose solution (0.3 and 0.06 M, respectively) was used as substrate to simulate typical compositions of sugar solutions obtained, for example, from the hydrolysis of inulin. Concentrations of the reagents and formed HMF were monitored at fixed reaction temperature (120 °C) at various reaction times, randomly modifying the contact time (in the interval 6–15 min· $g^{-1}$·$mL^{-1}$) during the reaction (Figures S3 and S4).

The obtained results (Figure S3) indicated that the reaction proceeded with high deactivation of all the catalysts because any clear decrease of initial substrate concentration or increase of formed HMF could not be observed with increasing contact time. In general, deactivation could be scarcely observed within the first 10 h of reaction and became clear over longer reaction times. NbO was clearly more deactivated than the NbO:NbP mixtures and NbP. During the course of the reaction, the trend of the fructose/glucose ratio first decreased, and then it increased again as deactivation limited the course of the reaction (Figure S4) Only on NbP, the fructose/glucose ratio maintained a constant value of ca. 1, even for a long reaction time.

Despite the limitation of the reaction extent caused by deactivation, some interesting information has been deduced, and it is shown in in Table 3. By comparing the kinetic constant values calculated at two different contact times (Table 3), it appears that the deactivation extent of catalysts was associated with the sample *effective* acidity and in particular with the LAS/BAS ratio and it decreased according to the following sequence: NbO >> NbO:NbP(3:1) ~ NbO:NbP(1:1) > NbP.

**Table 3.** Kinetic coefficients for the catalytic dehydration of glucose/fructose in water/ isopropanol solvent.

| Samples | Initial Velocity [a] ($h^{-1}$) | $k_{120}$ [b] ($min^{-1}$·mL $g_{cat}$) | |
|---|---|---|---|
| | | $\tau$ = 6 min $mL^{-1}$ $g_{cat}$ $^{-1}$ | $\tau$ = 15 min $mL^{-1}$ $g_{cat}$ $^{-1}$ |
| NbO | 0.27 | 0.10 | 0.011 |
| NbO:NbP(3:1) | 0.31 | 0.13 | 0.017 |
| NbO:NbP(1:1) | 0.38 | 0.20 | 0.021 |
| NbP | 0.36 | 0.20 | 0.061 |

[a] Calculated at t = 0.5 h by dividing the amount of converted reagents by the product of reaction time *per* the effective acid site density (in $H_2O$/2-propanol) and *per* the mass of catalyst. [b] kinetic constant calculated at 120 °C as reported in the Experimental section.

The initial velocity values have been calculated to individuate an activity ranking among samples, excluding the effects of deactivation (Table 3). The velocity values increased with the P-concentration on the samples; NbO:NbP(1:1) was as active as NbP.

The kinetic constants ($k_{120}$) at two different contact times, 6 min $mL^{-1}$ $g_{cat}$ $^{-1}$ and 15 min $mL^{-1}$ $g_{cat}$ $^{-1}$ (correspondent to initial and final reaction times), have also been calculated, as reported in the Materials and Methods section. At each contact time, the activity ranking already determined was validated, with NbP being the most active catalyst. The much lower values of $k_{120}$ at 15 min $mL^{-1}$ $g_{cat}$ $^{-1}$ than at 6 min $mL^{-1}$ $g_{cat}$ $^{-1}$ confirmed the catalyst deactivation, which governed the reaction.

## 3. Discussion

All the results obtained from our study and the targeted characterization experiments confirmed the water tolerance of LAS and BAS in the well-known NbO and NbP acid solids. It is then possible to affirm that a judicious addition of phosphate to NbO allowed tuning the nature of surface acid sites, progressively modifying the prevalent Lewis feature of NbO towards the prevalent Brønsted nature of NbP. Figure 7 shows the whole situation of the sample acidity in terms of quantity, nature, and strength of the acid sites of NbO, NbP, and related mixture samples, making a merge of the volumetric acid-base titration

measurements and the spectroscopically determined LAS/BAS ratios. The intrinsic and effective total numbers of the acid sites of each sample are not too different, thanks to the good water-tolerance of the acid sites of NbO and NbP. From Figure 7 emerges the good water-tolerance of both LAS and BAS of the two Nb-containing samples.

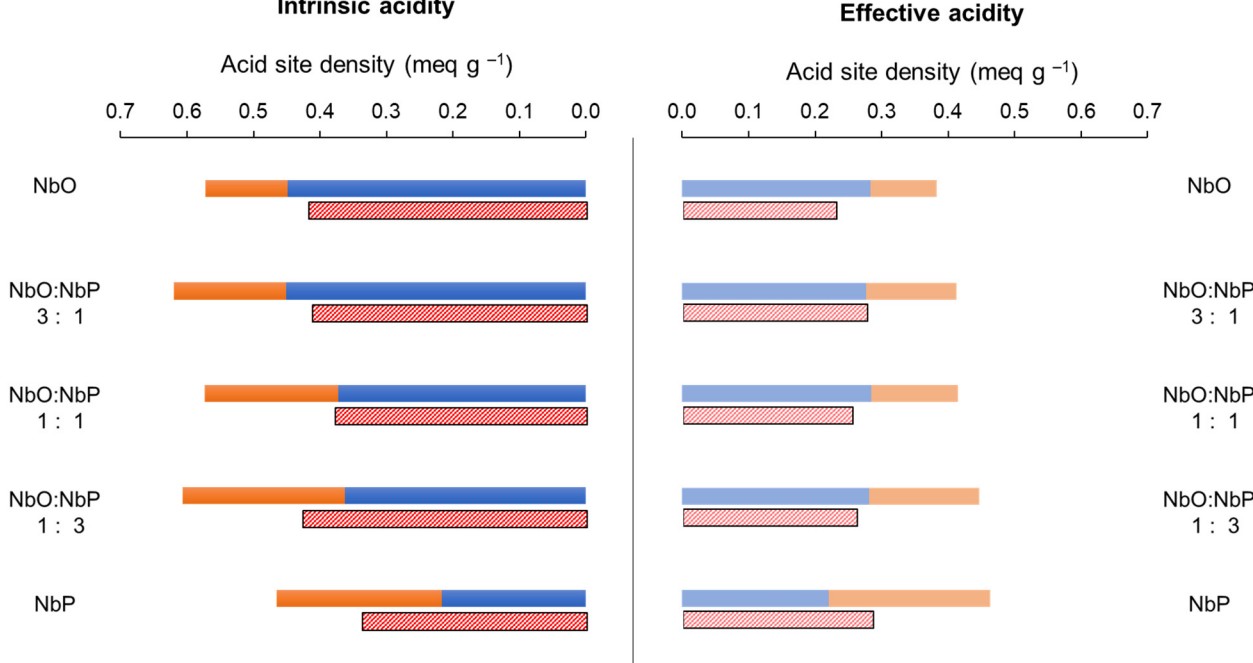

**Figure 7.** Surface acid site density of the studied samples: LAS and BAS determined by combining results from solid-liquid phase titrations with PEA (see Table S2) and FT-IR adsorption with pyridine (PY) (from desorption at 150 °C). Solvent, cyclohexane for intrinsic acidity, and water/isopropanol mixture for effective acidity measurements.

The catalytic effect of such gradual phosphate enrichment and Brønsted acidity enhancement has been investigated, unveiling the positive impact on the catalyst performance in BAS-promoted reactions.

In detail, the inulin depolymerization extent took advantage of the modification of surface acidity and experimented with an exponentially increasing trend with the P surface concentration, which reflected the exponential rise of both intrinsic and effective number of BAS.

Phosphate groups also improved the catalytic performances in the fructose/glucose dehydration by curbing the catalyst deactivation. A comparative view of the activity and stability of all the catalysts against contact time is shown in Figure 8. The fructose/glucose conversion was expected to have an increasing trend with the contact time, but more or less pronounced decreasing trends have been observed. Only NbP was able to maintain a good conversion throughout the reaction time (approx. 30 h), even if the deactivation caused by the formation of *humins* prevented the expected trend from being observed. The two mixtures, NbO:NbP(3:1) and NbO:NbP(1:1), showed a low decrease in conversion during the first part of the reaction (up to about 15 h of time) before the observed deep conversion drop. Furthermore, the initially observed conversion was higher on NbO:NbP(1:1) than on NbO:NbP(3:1). These observations confirm the positive role of surface phosphate groups (and Bronsted acidity) on the dehydration reaction in terms of activity and stability. In particular, the lower the surface P concentration on the samples, the more pronounced the activity decay.

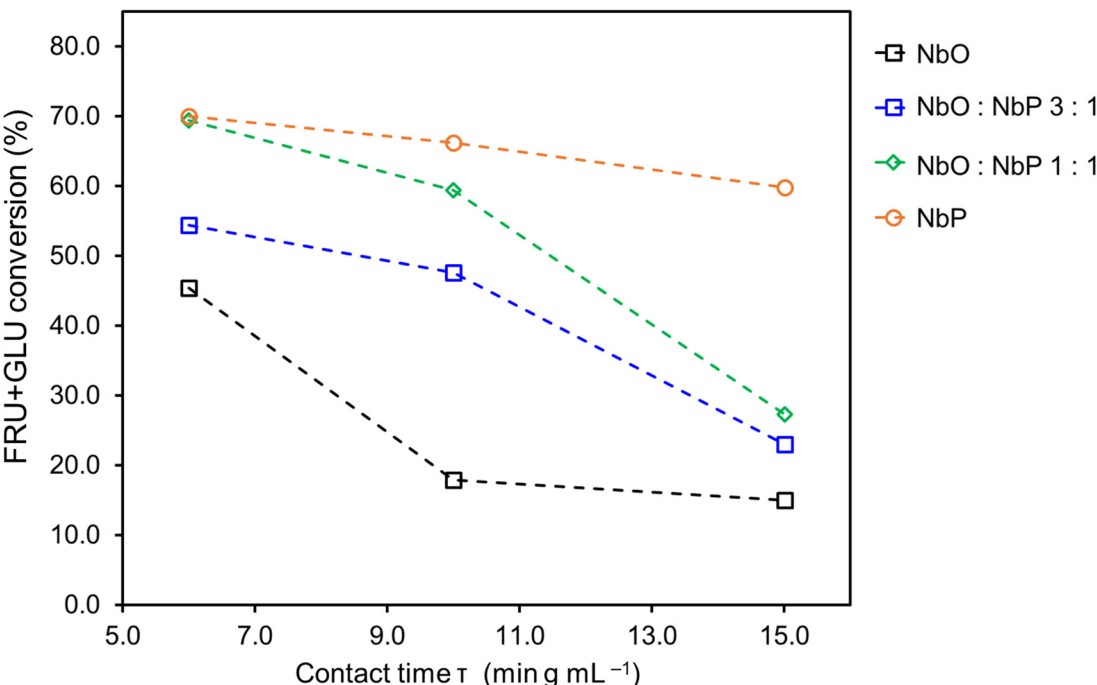

**Figure 8.** Trend of the conversion as a function of contact time in the continuous-flow reaction of glucose/fructose dehydration reaction (120 °C).

Future studies will be devoted to exploring the effect of phosphate enrichment in other reactions, requiring a controlled balance of LAS and BAS.

## 4. Materials and Methods

### 4.1. Materials

$Nb_2O_5 \cdot nH_2O$ (NbO, with nominal composition 80 wt% $Nb_2O_5$ and 20 wt% $H_2O$, code AD/5260) and $NbOPO_4 \cdot nH_2O$ (NbP, nominally composed by 66.7 wt% $Nb_2O_5$, 15.9 wt% $P_2O_5$, 2.1 wt% $K_2O$ and 15.3 wt% $H_2O$, code AD/5261) powder samples were kindly supplied from Companhia Brasileira de Metalurgia e Mineração (CBMM, Brazil).

All the materials and reagents used with indication of their source and purity have been detailed in Appendix A.

### 4.2. Catalyst Preparation and Characterization

Three physical mixtures with NbO:NbP mass ratios of 1:3, 1:1, and 3:1 were prepared in a ball mill equipment (Retsch, Mixer Mill MM 200, Haan, Germany) to guarantee the uniformity of the powders before any use. Samples were shaken for 30 min with a frequency of 10 Hz. The elemental composition and the homogeneity were assessed by energy-dispersive X-ray spectroscopy (EDX) (JEOL JSM-5500LV Scanning Electron Microscope, accelerating voltage max. 30 kV, magnification 5000×, EDS FWHM Resolution 128 eV, Tokyo, Japan).

Experiments were carried out aimed at stabilizing the phosphorus content of NbP to avoid leaching of P during the catalytic reactions that take place in the solid–liquid phase (water or water solution: isopropanol). The leaching of P can be attributed to the excess of phosphate groups resulting from the process of obtaining NbP (as indicated by CBMM). The experiments were performed as follows: 1.4 g of NbP was suspended in deionized water (20 mL) in a sealed tube put in an autoclave at 120 °C for a maximum of 2 h under 1.5–2 bar. Then, the solution was filtered and UV-vis spectrum was measured on the collected waters. The amount of phosphate present in the water samples was determined by the molybdenum blue phosphorous method [67]; the resulting blue complex gave a

signal at 820–830 nm. The experiments were repeated until there was no evidence of phosphate in the solution.

X-ray photoelectron spectroscopy analysis was performed on an M-PROBE Surface Science Spectrometer (Surface Science Instruments SSI, Mountain View, CA, USA), with an Al (Ka) source and a spot size from 150 μm to 1 mm in diameter. The applied voltage was 10 V at a vacuum of $10^{-7}$–$10^{-8}$ Torr. The survey scans were acquired in the binding energy range 0–1100 eV, using a spot size of 800 μm. The energy resolution was 4 eV and the scan rate was 1 eV per step. ESCA Hawk Software was employed for spectra elaboration.

### 4.3. Characterization of Surface Acidity

4.3.1. Acid Site Intrinsic Nature and Water Tolerance Properties

Lewis (LAS) and Brønsted (BAS) acid sites were determined by Fourier Transform Infrared Spectroscopy (FT-IR) (BioRad FTS-60A, Krefeld, Germany) using pyridine as a probe molecule under dry conditions. Powder samples (ca. 0.01–0.02 g) were pressed into a self-supporting wafer (12 mm of diameter), put in a glass IR cell with $CaF_2$ windows, and pretreated for 2 h at 150 °C under air, followed by outgassing for 30 min in high vacuum. Then, the pyridine adsorption was carried out by contacting the sample with the probe molecule vapors at room temperature for 10 min. The pyridine desorption was successively monitored by evacuating the sample for 30 min in a high vacuum at increasing temperatures (i.e., RT, 50, 100, 150, 200, and 250 °C) and cooling to room temperature between each step before recording the spectrum.

The concentrations of BAS and LAS were determined by integration of the peaks at 1540 and 1448 $cm^{-1}$, respectively, of the spectra collected after outgassing at 150 °C, according to Equation (1), proposed by Emeis [68]:

$$q_i = \left( A_i \pi R^2 \right) \left( w \varepsilon_i \right)^{-1} \tag{1}$$

where R (cm) is the radius of the sample wafer, w (mg) is the sample weight, and A is the integrated area of the characteristic bands of LAS or BAS, obtained after baseline correction. For the extinction coefficients, $\varepsilon$, of pyridine in interaction with BAS and LAS values from Emeis [68] were adopted ($\varepsilon_{BAS} = 1.67 \pm 0.12$ cm·μmol$^{-1}$ and $\varepsilon_{LAS}$ 2.22 $\pm$ 0.21 cm·μmol$^{-1}$, respectively).

The investigations of the water tolerance acid properties of the catalysts were carried out under the same experimental conditions. For this purpose, instead of using pyridine vapors, a $1 \times 10^{-3}$ M pyridine aqueous solution was dropped on self-supporting wafers under argon flow. The obtained wafers with pyridine/water co-adsorbed were measured by FT-IR monitoring pyridine desorption at 150 °C to quantify the amount of BAS and LAS.

4.3.2. Liquid–Solid Acid Site Titration for the Determination of Intrinsic and Effective Acidity

The amount of acid sites and their strength were determined by liquid–solid phase titration by using 2-phenylethylamine (PEA, pK$_a$ = 9.83) as basic probe. The titrations were carried out on powder samples (around 80–100 mg, 75–180 μm) at 30 $\pm$ 0.1 °C in cyclohexane, an apolar/aprotic solvent, for the *intrinsic* acidity (I.A.) and in water:isopropanol mixture (80:20 *v/v*), a polar/protic solvent, for the *effective* acidity (E.A.). Analyses were performed in a modified liquid-chromatographic line operating in a recirculating mode [19,46,69,70].

The collected isotherms were interpreted following the Langmuir model (Equation (2)):

$$\frac{PEA_{ads}}{PEA_{ads,max}} \frac{b_{ads}[PEA]_{eq}}{\left(1 + b_{ads}[PEA]_{eq}\right)} \tag{2}$$

where $PEA_{ads}$ and $PEA_{ads,max}$ indicate the amount of PEA adsorbed and the maximum amount of PEA adsorbed by the surface, $[PEA]_{eq}$ indicates the PEA concentration in solu-

tion at equilibrium, and $b_{ads}$ is the Langmuir constant. From the conventional linearized equation, reporting $[PEA]_{eq}/[PEA]_{ads}$ vs. $[PEA]_{eq}$, the values of $PEA_{ads,max}$ (mmol·g$^{-1}$) could be obtained. Assuming a 1:1 stoichiometry for the PEA adsorption on the acid site, the value of $PEA_{ads,max}$ of the I run isotherm gives the total number of acid sites, while the value of $PEA_{ads,max}$ obtained from the II run isotherm corresponded to the number of weak acidic sites. The number of strong acid sites was obtained by the difference between the number of total and weak sites.

All the details on the operative and interpretative procedures can be found in Appendix A.

### 4.4. Catalytic Experiments and Analysis

4.4.1. Catalytic Tests of Inulin Hydrolysis

The catalytic hydrolysis of inulin was performed in aqueous phase in a glass slurry batch reactor (Syrris, Atlas, Royston, UK) with magnetic stirrer at a constant rate of 800 rpm under increasing temperature from 50 to 90 °C (0.1 °C min$^{-1}$, for a total of 6 h of reaction).

Prior to the reaction, the catalyst sample (ca. 0.3 g) was treated at 120 °C for 16 h in an oven under air atmosphere. Inulin (ca. 1.5 g) was put into the reactor with 150 mL of water at 50 °C; after catalyst addition, the reaction started.

The reaction (Scheme 1) was followed by measuring the total reducing sugars by the classical colourimetric Nelson–Somogyi method. The blue complex formed was quantified at 520 nm. In parallel, the concentrations of glucose and fructose were determined by D-glucose/D-fructose Boehringer Mannheim enzymatic assay (Enzymatic BioAnalysis/Food Analysis by Roche-Biopharm).

The inulin conversion, that is, the extent of hydrolysis reaction, has been calculated as the ratio between the reducing sugars at any given time/temperature of reaction and the reducing sugars at complete inulin hydrolysis. The selectivity to fructose and glucose was evaluated as ratio between produced fructose or glucose and all the formed reducing sugars.

Details on the kinetic interpretation for the determination of activation parameters are reported in Appendix A.

4.4.2. Catalytic Tests of Fructose and Glucose Dehydration

The dehydration of fructose and glucose (0.3 and 0.06 M, respectively) to HMF was performed in a continuous reaction line equipped with a pump (HITACHI Merck L-6200 Intelligent Pump, Tokyo, Japan), stainless steel pre-heater and a tubular catalytic reactor (length 120 mm, i.d. 6 mm). The pre-heater and the reactor were assembled in an oven kept at a constant temperature (120 °C). The sample (1.5 g, sieved to 350–700 μm) was held in the middle of the reactor between two sand pillows (0.5 g each). Water:isopropanol (80:20 *v/v*) was used as the solvent. The monosaccharides solution was continuously fed into the catalytic bed reactor at different fixed flow rates (0.1 to 0.25 mL min$^{-1}$) to vary contact times (from 6.0 to 15.0 min·g$^{-1}$·mL$^{-1}$) during the reaction course. After each change of flow rate, at least 30 ml of solution was left to flow to ensure the stationary conditions. At least three analyses were performed at each contact time, averaging the results. The pressure in the reactor was kept between 6 and 10 bar by means of a micrometric valve at the end of the reaction line to guarantee the solvent in a liquid phase.

The products were analyzed in a high-performance liquid chromatographic (HPLC), consisting of injector (Waters U6K, Sesto San Giovanni, Italy), pump (Waters 510, Sesto San Giovanni, Italy), heater (Waters CHM, Sesto San Giovanni, Italy) for the column, and refractive index detector (Waters 410, Sesto San Giovanni, Italy). A Sugar-Pack I (300 × 6.5 mm, 10 μm particle size, Waters, Sesto San Giovanni, Italy) column operating at 90 °C and eluted with an aqueous solution of Ca–EDTA ($10^{-4}$ M) was used.

The kinetic constants ($k_T$) values can be obtained by the following equation:

$$-dC/d\tau = k_T\,C$$

where $\tau$ is the contact time in $\text{min}\cdot\text{mL}^{-1}\,\text{g}_{cat}^{-1}$. Assuming a first-order reaction [22]

$$k_T = 1/\tau \ln (C^\circ/C)$$

where $C^\circ$ and $C$ are the concentrations of the reagent, respectively, before and after the catalytic bed.

**Supplementary Materials:** The following are available online at https://www.mdpi.com/article/10.3390/catal11091077/s1, Figure S1: FT-IR spectra of the catalysts samples with pyridine vapor (A) and pyridine aqueous (B) after pyridine desorption at 150 °C. The baselines of all spectra have been corrected, Figure S2: Cumulative trend of the reducing sugars against inulin conversion, taken as an index of reaction course, observed on all the catalysts in the inulin hydrolysis reaction, Figure S3: Curve profiles obtained from the continuous-flow reaction of glucose/fructose dehydration collected at 120 °C: concentration of reagents (GLU+FRU) and HMF formed as a function of reaction time at different contact times: ● = 6 $\text{min}\cdot\text{g}^{-1}\cdot\text{mL}^{-1}$; ■ = 10 $\text{min}\cdot\text{g}^{-1}\cdot\text{mL}^{-1}$; ◊ = 15 $\text{min}\cdot\text{g}^{-1}\cdot\text{mL}^{-1}$; ▲ = 7.5 $\text{min}\cdot\text{g}^{-1}\cdot\text{mL}^{-1}$, Figure S4: Plot of the fructose to glucose ratio as a function of reaction time in the continuous-flow reaction of glucose/fructose dehydration at 120 °C at different contact times: ● = 6 $\text{min}\cdot\text{g}^{-1}\cdot\text{mL}^{-1}$; ■ = 10 $\text{min}\cdot\text{g}^{-1}\cdot\text{mL}^{-1}$; ◊ = 15 $\text{min}\cdot\text{g}^{-1}\cdot\text{mL}^{-1}$; ▲ = 7.5 $\text{min}\cdot\text{g}^{-1}\cdot\text{mL}^{-1}$, Table S1: EDX and XPS compositions of Nb-bases samples, Table S2: Determination of the nature and surface density of the acid sites of NbO and NbP solids and physical mixtures determined by FT-IR with pyridine (PY) adsorption in vapor phase with/without water and by solid-liquid phase titrations with phenyl-ethylamine PEA.

**Author Contributions:** Conceptualization, M.N.C., A.G. and P.C.; methodology, A.G. and P.C.; validation, A.G. and S.C.; formal analysis, S.C.; investigation, M.N.C. and F.B.; resources, A.G.; data curation, A.G. and S.C.; writing—original draft preparation, M.N.C. and S.C.; writing—review and editing, A.G. and S.C.; visualization, S.C.; supervision, A.G.; project administration, A.G.; funding acquisition, A.G. and R.F.T. All authors have read and agreed to the published version of the manuscript.

**Funding:** This research received no external funding.

**Data Availability Statement:** The data that support the findings of this study are available on demand but restrictions apply to the availability of these data, which were used for the current study and are part of other studies still in progress. Data are however available from the authors upon reasonable request and with permission of the corresponding author (A.G.).

**Acknowledgments:** The authors are grateful to CAPES/PDSE for providing the scholarship to Mariana Neves Catrinck (grant n° 88881.135387/2016-01), and to Companhia Brasileira de Metalurgia e Mineração (CBMM) for supplying the niobium-based catalysts.

**Conflicts of Interest:** The authors declare no conflict of interest.

## Appendix A

*Appendix A.1 Experimental Details*

Reagents and Materials

Cyclohexane (AnalaR, NORMAPUR, ACS, Reag. Ph. Eur. from VWR), 2-propanol (HPLC grade, 99.9% from Merck Sigma-Aldrich), water (HPLC grade, from VWR BDH Chemicals), 2-phenylethylamine (≥99%, from Sigma-Aldrich) were used for solid–liquid phase titrations.

Inulin (from chicory root, loss on drying ≤10%, free fructose, glucose, sucrose ≤5%) in powder form with a number-average degree of polymerization ($DP_n$) of 6 was purchased from Acros Organics.

Glucose (99.5%), D(-)-Fructose (≥99%) and 5-(Hydroxymethyl)-furfural (99+%) were supplied from Sigma-Aldrich.

Ammonium heptamolybdate tetrahydrate $(NH_4)_6MoO_{14}\cdot4H_2O$ (99.98% from Sigma-Aldrich), $H_2SO_4$ (98% from Sigma-Aldrich), and $Na_2HAsO_4\cdot7H_2O$ (99%, from Carlo Erba) were mixed to prepare Nelson reagent.

Somogyi reagent was prepared by mixing (i) sodium phosphate dibasic dodecahydrate $Na_2HPO_4$ $12H_2O$, (ii) potassium sodium tartrate tetrahydrate $COOK(CHOH)2COONa·4H_2O$ (99%, from Carlo Erba), (iii) sodium hydroxide solution 2 N (from Carlo Erba), (iv) sodium sulfate decahydrate $Na_2SO_4·10H_2O$ ($\geq$99%, from Sigma-Aldrich), and v) copper sulfate pentahydrate $CuSO_4·5H_2O$ (99%, from Sigma-Aldrich).

*Appendix A.2 Solid–Liquid Phase Titration of Surface Acid Sites by Phenylethylamine Probe*

A weighed amount (ca. 0.05 g) of sample previously crushed and sieved as 80–200 mesh particles was placed between two sand pillows into a stainless-steel tubular sample holder (2 mm i.d. and 13 cm of length). Prior to the measurements, the sample was thermally pre-treated at 150 °C overnight under flowing air (8 mL min$^{-1}$) and then evacuated and filled with the proper solvent (cyclohexane or water:isopropanol mixture). The sample holder, inserted in a glass jacket connected with a thermostatic water bath to maintain a constant temperature (30.0 ± 0.1 °C), replaced the conventional chromatographic column. A pump (Waters 510, Sesto San Giovanni, Italy) and a monochromatic UV detector (Waters 2487, $\lambda$ = 254 nm, Sesto San Giovanni, Italy) completed the analytical apparatus.

The acid sites were titrated by injecting successive doses (0.05 mL) of PEA solution (0.1 M in cyclohexane or water:isopropanol) into the line where the solution continuously circulated at a flow of 3 mL·min$^{-1}$ until adsorption equilibrium was achieved. Raw data obtained from the titration experiments were in the form of a step-chromatogram, consisting of a series of increasing steps, each one representing the attainment of the adsorption equilibrium. The titration was considered concluded when at least three consecutive steps possessed the same height, indicating the attainment of surface saturation. After the collection of the first adsorption isotherm of PEA on the fresh sample (I run), the pure solvent was pumped at a flow of 0.1 mL min$^{-1}$ through the sample overnight to remove the PEA feebly adsorbed on weak acid sites. Then, a new adsorption run of PEA on the same sample was repeated (II run).

Numerical interpretation of the PEA adsorption isotherms was carried out considering the Langmuir model equation.

*Appendix A.3 Kinetic Interpretation of Catalytic Data*

Inulin Hydrolysis

For inulin hydrolysis, the kinetic coefficients ($k_{Tm}$) were computed at the average temperature $T_m$ between two samplings according to Equation (A1):

$$k_{Tm} = (\Delta C / \Delta t)/C_m \qquad\qquad (A1)$$

where $\Delta t$ is the time interval between two samplings ($\Delta t$ = 30 min and $\Delta T$ = 5 °C); $\Delta C$ is the variation of bond concentration between two samplings; $C_m$ is the average bond concentration between two samplings.

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
