# Peer review of "Phosphate Enrichment of Niobium-Based Catalytic Surfaces in Relation to Reactions of Carbohydrate Biomass Conversion: The Case Studies of Inulin Hydrolysis and Fructose Dehydration"

_catalysts, doi:10.3390/catal11091077_

Round 1

Reviewer 1 Report

Dear Authors, in my opinion, the article presents well-performed and innovative research.

Some minor remarks:

How would you rate the accuracy of XPS and EDX when determining the concentration of elements (Figure 1)?

The title of this article will attract many readers interested in practical carbohydrate biomass conversion, including cellulosic biomass, but the content relates to the isolated inulin only. Maybe the title should be more detailed?

Author Response

Dear Reviewer,

Thank you for giving us the opportunity to submit a revised version of our manuscript for publication in Catalysts. We appreciate the time and efforts of Editors and Reviewers to providing feedback on our manuscript and are grateful to them for the appreciative comments and valuable inputs for the improvement of our paper. We have modified the text according to the suggestions made by the reviewers. The performed changes are marked up within the manuscript using the “Track Changes” function of MS Word. Please find below, in red colour, our point-by-point response to the reviewer comments/questions.

Reviewers' Comments to the Authors: Reviewer 1 – Round 1

Dear Authors, in my opinion, the article presents well-performed and innovative research. We are grateful to the Reviewer for the positive feedback on the quality of our research. Some minor remarks: How would you rate the accuracy of XPS and EDX when determining the concentration of elements (Figure 1)? The accuracy in XPS measurements is influenced by various parameters, among these the most important are: i) accuracy of relative sensitivity factors (RSF), ii) signal-to-noise ratio, iii) background type, and nature of the element. Thus, XPS atomic concentrations are typically determined with an accuracy of ca. 10% (see A.G. Shard, Practical guides for x-ray photoelectron spectroscopy: Quantitative XPS, J. Vac. Sci. Technol. A 38, 041201 (2020); doi: 10.1116/1.5141395; D. Briggs and M.P. Seah, Eds., Practical Surface Analysis, Vol. 1, Wiley, Chichester, 1990). The accuracy of EDX as well depends on several parameters, including sample morphology, nature of the element, instrumental errors. For the elements analyzed in our study (Nb, P and O), EDX standardless quantitative measurements performed on pelletized samples were affected by an estimated relative error of ca. 2.5 - 7% (based on comparative analysis with reference standards).

The title of this article will attract many readers interested in practical carbohydrate biomass conversion, including cellulosic biomass, but the content relates to the isolated inulin only. Maybe the title should be more detailed?

Thank you for the suggestion. The manuscript title has been now modified by additionally specifying the two reactions this study focuses on. The following new title proposed is: “Phosphate enrichment of niobium based catalytic surfaces in relation to reactions of carbohydrate biomass conversion: the case studies of inulin hydrolysis and fructose dehydration”

Reviewer 2 Report

Evaluation of manuscript entitled:

Phosphate enrichment of niobium based catalytic surfaces in relation to reactions of carbohydrate biomass conversion

The aim of this manuscript was to demonstrate the role of phosphate groups in the physical mixtures of Nb2O5∙nH2O and NbOPO4 used in two reactions for the conversion of carbohydrate biomass: hydrolysis of polyfructane and dehydration of fructose/glucose to 5-HMF. First, the characterization of the solid mixtures with a focus on acidic properties was thoroughly performed, followed by the catalytical testing of the mixtures. The amount of phosphate groups in the solid mixtures dominated the acid and catalytic properties. The authors have found that Lewis acidity decreased with the amount of NbP in the composition, while total acidity of the samples were practically unchanged. Inulin depolymerization using catalyst rich in BAS and NbP showed the best hydrolysis activity. However, during fructose/glucose dehydration significant catalyst dehydration was found. The work presents an interesting work; however, some smaller issues and questions should be addressed before publication:

Major issues/questions:

  1. Why did the authors choose 120 °C for the fructose dehydration? Does the temperature influence the amount of side-product (humin) formation? Could the effect of temperature on the product conversion/catalyst deactivation be more significant than the residence time?
  2. Were there repeated experiments that show the reproducibility of the solid mixture preparation?
  3. Did the authors perform parallel experiments for the batch reaction with inulin?
  4. Why is it necessary to treat the catalyst at 120 °C for 16 h in the oven before the reaction?
  5. Have the authors tried the dehydration reaction with the crude mixture gained from the depolymerization?
  6. Is it possible to perform the two steps in one reaction set-up?

Minor issues:

  1. In the abstract, use of abbreviations should be avoided.
  2. An additional reference might be included in the text: Kang et al. Sustainable production of fuels and chemicals from biomass over niobium based catalysts: A review (https://doi.org/10.1016/j.cattod.2020.10.029).
  3. Based on the SI, potassium is an impurity in the samples. This should be added as a note to Table 1 in the manuscript as well.
  4. Regarding the paragraph about inulin (2.3.1 Catalytic inulin hydrolysis) some references should be added to provide further information for the reader if they are interested.
  5. What is considered as a “sufficient” fructose/glucose ratio for the reaction?
  6. Some minor issues regarding the used language are collected in the attached pdf. I would also suggest to check the spelling in the SI.

All things considered, this manuscript addresses an important research topic, however, before publication some minor issues (mostly to satisfy the interest of the reviewer) should be addressed. Therefore, I suggest the major revision of this article in Catalysts.

Author Response

Answers to review 2 are sent by an attached file.

Round 2

Reviewer 2 Report

Thank you for your detailed answers, and congratulations on the interesting work.